behaviour/psychology

violent media, binocular rivalry, fans of violent music, implicit process, violent imagery

**Author for correspondence:**
Yanan Sun
e-mail: yanan.sun@mq.edu.au

†These authors contributed equally to this study.

# Implicit violent imagery processing among fans and non-fans of music with violent themes

Yanan Sun[1,2,3,†], Xuejing Lu[4,5,†], Mark Williams[2,3] and William Forde Thompson[1,3]

[1]Department of Psychology, and [2]Department of Cognitive Science, Macquarie University, New South Wales 2109, Australia
[3]ARC Centre of Excellence in Cognition and Its Disorders, New South Wales 2109, Australia
[4]CAS Key Laboratory of Mental Health, Institute of Psychology, Beijing 100101, People's Republic of China
[5]Department of Psychology, University of Chinese Academy of Sciences, Beijing 100049, People's Republic of China

 YS, 0000-0002-2560-2791; XL, 0000-0001-7586-5576; MW, 0000-0002-3897-5167; WFT, 0000-0002-4256-1338

It is suggested that long-term exposure to violent media may decrease sensitivity to depictions of violence. However, it is unknown whether persistent exposure to music with violent themes affects implicit violent imagery processing. Using a binocular rivalry paradigm, we investigated whether the presence of violent music influences conscious awareness of violent imagery among fans and non-fans of such music. Thirty-two fans and 48 non-fans participated in the study. Violent and neutral pictures were simultaneously presented one to each eye, and participants indicated which picture they perceived (i.e. violent percept, neutral percept or blend of two) via key presses, while they heard Western popular music with lyrics that expressed happiness or Western extreme metal music with lyrics that expressed violence. We found both fans and non-fans of violent music exhibited a general negativity bias for violent imagery over neutral imagery regardless of the music genres. For non-fans, this bias was stronger while listening to music that expressed violence than while listening to music that expressed happiness. For fans of violent music, however, the bias was the same while listening to music that expressed either violence or happiness. We discussed these results in view of current debates on the impact of violent media.

## 1. Introduction

Violent media refers to 'those that depict intentional attempts by individuals to inflict harm on others' [1, p. 354]. It has been

suggested that exposure to violent media causes significant increases in aggressive thoughts, aggressive behaviours, angry feelings and physiological arousal levels (e.g. [1–5]). There are also concerns that exposure to violent media may lead to desensitization—the attenuation or elimination of physiological, cognitive, emotional and behavioural responses to subsequent scenes of violence, such as decreased helpful behaviours, a reduction in sympathy for a violence victim and/or a toleration for 'real-life' violence (e.g. [2,6–13]; but see [14]).

Most research on the psychosocial effects of exposure to violent media has focused on screen-based media, such as television, movies and video games. There is far less research on the consequences of exposure to music with violent themes. Some genres of music, including but not limited to extreme metal, rap and hip hop, are often, though certainly not always, characterized by aggressive sounds and violent lyrics. As such, many researchers are interested in whether exposure to music with violent themes leads to aggressive thoughts, feelings, emotions and behaviours (e.g. [5,15–20]; but see [21,22]). To date, it has been difficult to infer causal links between long-term exposure to violent music and aggressive or violent behaviours, because research has often relied on correlational evidence. For example, it has been reported that preference of heavy metal music is correlated with reckless behaviours [23–25], low academic performance [26] and mental problems such as depression and anxiety [27]. However, it is difficult to conclude that exposure to heavy metal music played a causal role in such outcomes and should therefore be avoided. Indeed, some research has highlighted the psychosocial benefits of listening to extreme metal music, whereby some fans use extreme music to process feelings of anger, and to relax [28].

Such contrasting findings highlight the need for more research on the effects of listening to music with violent themes. Such research would also address abiding public concerns about media violence, which have persisted ever since the US Surgeon General called for research on the topic [29]. One potential effect of exposure to media violence is desensitization, defined as a reduction in emotion-related reactivity to violence. Exposure to violent media, including violent music, may reduce a listener's threshold for experiencing distress and alarm when witnessing violence activities [8]. The present study used a binocular rivalry paradigm to investigate this possibility for fans of extreme metal music with violent lyrics. Binocular rivalry occurs when perceptually distinct images are presented to each eye (e.g. a face to one eye and a house to the other eye) and compete for perceptual dominance. Typically, visual input from one eye is dominant (and seen), while the other image is suppressed (and remains unseen). However, individuals frequently experience the two images as alternating over time. By measuring the amount of time that each image is dominant (or suppressed), it is possible to determine which visual input the brain is prioritizing for conscious experience.

Neuroimaging and psychophysiological studies indicate that rivalry depends on competitive interaction at multiple neural sites and at different levels of processing (see [30] for a review). Dominance is influenced by physical characteristics of visual stimuli (i.e. bottom-up processing), such as luminance [31], contrast [32] and configural properties [33]. Central to the current investigation, highly emotional images tend to dominate over affectively neutral images in binocular rivalry paradigms [34–37]. Voluntary control and controlled attention do not influence which image is consciously seen [38], but some forms of top-down processing can influence the contents of consciousness. For example, the affective state of perceivers can impact upon binocular rivalry results, with increased awareness of emotional images that are congruent with the affective state of participants [39].

Our music stimuli were classified as violent or happy based on properties of the music, rather than the experiences of participants. The emotional responses and meaning that individual listeners derive from a particular piece of music will depend on their current psychological states, goals, personality and sociocultural background [40], and need not align with emotions that are expressed by the music [41]. For this reason, media violence researchers invariably define 'violence' according to societal and legal definitions rather than according to the emotional or aesthetic responses of enthusiasts of such media. Fans of video games that explicitly depict murder may experience empowerment and joy from playing the game, and may even derive psychosocial and cognitive benefits, but such positive experiences and outcomes should not obscure the fact that the game portrays violent acts (for relevant discussions, see [42,43]).

For our stimuli, we selected contrasting songs for which both sonic and lyric contents were congruent in emotional connotation. Thus, our exemplar of violent music contained both explicitly violent lyrics (i.e. lyrics that depicted the harm or murder of an individual without their consent) and sonic features that have been confirmed empirically to be associated with aggression (e.g. sounds with the qualities of growling, shouting and screaming) [44–46]. In particular, biological signals of aggression are known to be communicated by acoustic attributes such as fundamental frequency [47–49], intensity

level [50] and pace (inter-onset interval [51]). As control stimuli with contrasting connotations, music expressing positive emotions was also included, which has explicitly joyful lyrics and musical features that are associated with positive emotions, such as major mode, high mean pitch and consonant harmonies [52–57].

Two groups were tested: fans of aggressive or violent music, and a group of participants who explicitly reported that they were not fans of such music. Groups were initially recruited by identifying participants who either did or did not enjoy listening to heavy and death metal music. However, because these genres do not always express violence or aggression in the lyrics, participants were further required to indicate how often they listen to music with *violent themes*. This step ensured that there were large and reliable differences in the two groups with respect to their enjoyment of music with violent lyrics, as well as differences in their reported enjoyment of music with aggressive acoustic attributes. Participants listened to each musical stimulus while violent and affectively neutral images were presented to them binocularly, resulting in competition or 'rivalry' between the two images. Consistent with past research, violent imagery should generally dominate consciousness over neutral imagery. Moreover, for most people, this tendency to perceive violent images should occur earlier and for longer durations while listening to violent music than while listening to non-violent music, reflecting a 'congruence effect' in which emotions experienced while listening to music reinforce the emotions expressed in images [39]. Non-fans of violent music should exhibit this congruence effect because according to previous research, non-fans of death metal music typically have very negative emotional responses to this music [43] but should have positive emotional responses to upbeat Western popular songs with joyful lyrics written in a major key [58,59]. By contrast, fans of death metal music should experience positive emotions while listening to their preferred music. Thus, their bias for processing violent images may not be differentially affected by the two types of music.

## 2. Methods

### 2.1. Participants

Thirty-two (24 males) self-declared fans of death metal or heavy metal music and 48 (24 males) non-fans of such music participated in the study. Participants were Macquarie University students ranging in age from 18 to 35 years ($\bar{X} = 21.22$ years, s.e. $= 0.38$ years). Group membership was determined by responses to the question: are you a fan of death metal or heavy metal music (yes/no)? Participants were then further asked how long on average they spend each week listening to music with *violent themes*. The fan group reported spending significantly more time listening to this kind of music than the non-fans (fans: $\bar{X} = 7.37$ h, s.e. $= 2.23$ h; non-fans: $\bar{X} = 0.38$ h, s.e. $= 0.23$ h; $t_{corrected}$ (1,78) $= 3.11$, $p = 0.004$). This explicitly indicates that metal fans in our study frequently listened to and enjoyed metal music with *violent themes*. All participants reported normal hearing and normal or corrected-to-normal vision. None reported any auditory, neurological or psychiatric disorder.

### 2.2. Materials

#### 2.2.1. Auditory stimuli

Two songs that contrasted in expressed emotion were selected for the investigation. The first piece, 'Eaten' by Bloodbath, was selected because it has explicitly violent lyrics that are accompanied by music with acoustic qualities that are characteristics of biological signals of aggression (e.g. nonlinearities, low mean pitch level, growling or screaming vocalizations). The second piece, 'Happy' by Pharrell Williams, was selected because it depicts prosocial and joyous lyrics accompanied by musical qualities with a positive emotional connotation (e.g. conventional song structure, major tonality, consonant sonorities and high mean pitch level).

Although the two songs contained many distinct features, we attempted to control for simple differences in arousal arising from loudness (intensity) and tempo [60–63]. This procedure was carried out in order to reduce the potential impact of physiological arousal on binocular rivalry processing [64], and thereby increase the potential impact of the contrasting emotional connotations of the two songs. To this end, the audios of the two songs were matched on the root-mean-square of intensity, and their average tempo set to 148 bpm as measured by MIR toolbox [65]. Thus, any differences in the impact of songs on binocular rivalry could not be attributed to simple differences in loudness or

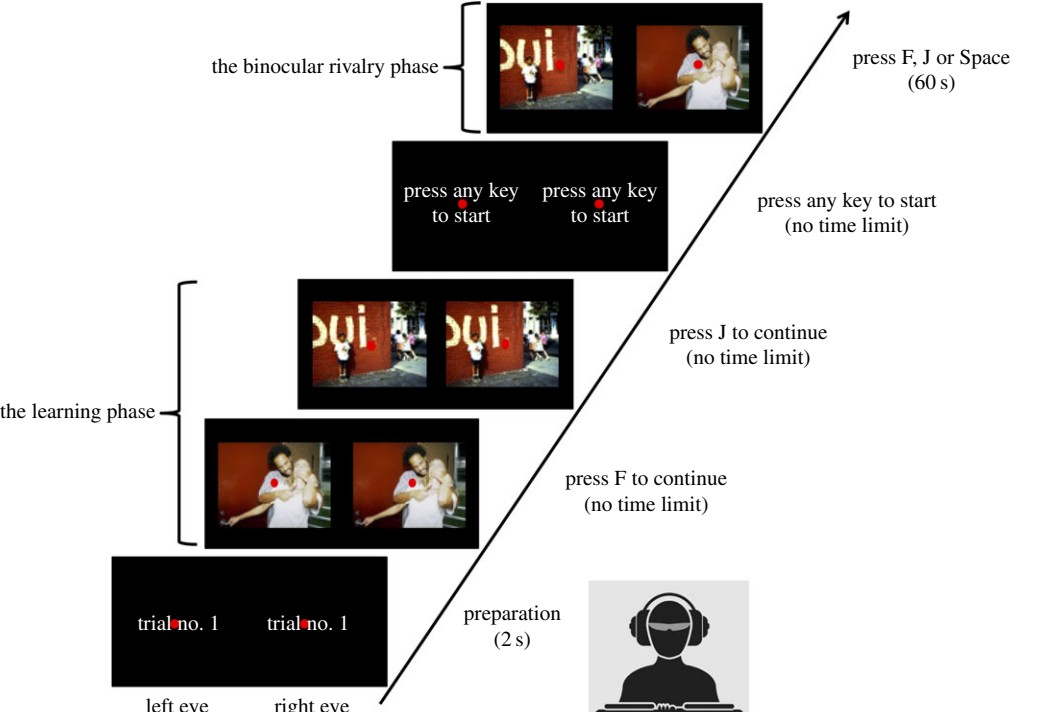

**Figure 1.** The procedure of the binocular rivalry experiment.

tempo, but would most probably arise from the striking differences in the emotional connotations of the two songs. The auditory stimuli were presented at a comfortable hearing level (approx. 65 dB SPL) through Sennheiser HD 280 Pro headphones.

### 2.2.2. Visual stimuli

Eleven violent–neutral image pairs were created for the binocular rivalry task. All images were selected from the International Affective Picture System[1] (IAPS [66]). For each pair, two pictures were matched on physical characteristics such as luminance and contrast using Adobe Photoshop CS4 Extended (v. 11.0.2) colour match tool. To ensure the pictures were big enough to be clearly perceived but small enough to minimize blended percept, all selected pictures were rescaled to $120 \times 160$ pixels. They were presented using Psychtoolbox (v. 3.0) implemented in Matlab (v. 8.6) running on an iMac (OS 10.11.6) with an Oculus Rift Development Kit 2 (Oculus VR, LLC) as a stereoscope (resolution: $960 \times 1080$ pixels per eye; refresh rate: 75 Hz). To help participants focus on the visual stimuli, a small red dot was displayed in the centre of each picture and served as the fixation point.

### 2.3. Procedure

As shown in figure 1, each trial began with a learning phase, followed by a binocular rivalry phase. During the learning phase, participants viewed two pictures of a pair one after another. In this phase, each picture was displayed to both eyes. Participants were then asked to press any key to start the binocular rivalry phase. In this phase, participants were presented with both pictures simultaneously, with the neutral picture presented to one eye and the violent picture presented to the other eye. On each trial, the two images were displayed for 60 s. Within each block of trials, the inter-trial interval was 2 s. Participants were instructed to focus on the central fixation point and keep their fingers on the response keys (F', 'J' and 'SPACE') all times during the task. Half of the participants were told to press 'F' when they perceived the first picture that they have learned during the learning phase, and

[1]The practice session used the following pairs of IAPS pictures: 6312–5471, 6350–7175 and 6550–2840. The test session used the following pairs of IAPS pictures: 2683–2521, 2691–7036, 6313–2272, 6315–2026, 6520–2273, 6560–2309, 6821–7033 and 9414–7037. Based on the original IAPS affective ratings, the mean valence rating is 2.31 for violent pictures and 5.06 for neutral pictures, whereas the mean arousal rating is 6.41 for violent pictures and 3.76 for neutral pictures.

press 'J' when they perceived the second picture. The other half were told to use reversed response keys. All participants were asked to press 'SPACE' if they saw both pictures or a blend of the two.

Three violent–neutral image pairs were used in the practice block and the other eight image pairs were used in the test blocks. Given that the picture presented to the dominant eye is more likely to be seen, each violent–neutral image pair was presented twice at random positions in the sequence of trials, where the violent was present to the left (right) eye in one trial and to the right (left) eye in the repeated trial. In total, there were six trials for the practice block and 16 trials for the test block. All participants were given a practice block with no music at the beginning of the experiment, and then asked to take two test blocks. The song Happy was played throughout one test block, and the song Eaten was played throughout the other test block. The two test blocks consisted of the same images but different music. Half of the participants were presented with the happy-music block first, and the other half were presented with the violent-music block first. The order of trials within the practice and test blocks was randomly determined for each participant independently. Participants were encouraged to take a break between the two blocks. After each test block, participants were required to rate the song they had just heard in terms of its valence (1, unpleasant; 7, pleasant) and arousal (1, relaxing; 7, arousing) based on their own feelings.

## 2.4. Data analysis

For the binocular rivalry task, two indexes were calculated for each participant. One was the proportion of the initial percepts (violent percept or neutral percept) out of the 16 trials, which was considered as a more objective measurement [67]. Another one was the mean percept duration across trials for violent pictures and neutral pictures, respectively. Blended percepts were excluded from the analysis. Each index (i.e. mean percept duration and proportion of initial percept) was analysed using a repeated-measures ANOVA with a between-subjects factor of Group (non-fans versus fans) and two within-subjects factors of Music (violent versus happy) and Picture (violent versus neutral). We also examined ratings of the music for valence and arousal separately using a repeated-measures ANOVA with Group (fans versus non-fans) as the between-subjects factor and Music (violent versus happy) as the within-subjects factor. For all analyses, it was agreed in advance that if the interaction was significant, a test of simple effect would be conducted to explore the interaction. Generalized $\eta^2$ was reported as effect sizes [68].

## 3. Results

For the index of the proportion of initial percept, the repeated-measures ANOVA revealed a significant main effect of Picture ($F_{1,78} = 30.01$, $p < 0.001$, $\eta^2_G = 0.086$). The interaction between Group and Picture was not significant ($p > 0.8$), which means both fans and non-fans of violent music perceived the violent pictures more frequently than the neutral pictures regardless of the music (fans: for violent pictures, $\bar{X} = 0.41$, s.e. $= 0.03$; for neutral pictures, $\bar{X} = 0.31$, s.e. $= 0.02$; non-fans: for violent pictures, $\bar{X} = 0.41$, s.e. $= 0.02$; for neutral pictures, $\bar{X} = 0.30$, s.e. $= 0.02$). We also found a significant interaction effect among Music, Picture and Group ($F_{1,78} = 4.85$, $p = 0.031$, $\eta^2_G = 0.010$). No other effects were significant ($ps > 0.1$).

To explore the significant interaction, separate analyses were conducted for each picture type. For the violent picture condition, there was a significant interaction between Music and Group ($F_{1,78} = 4.45$, $p = 0.038$, $\eta^2_G = 0.014$). A simple effect test showed that non-fans reported seeing violent pictures significantly more often while listening to the song Eaten than while listening to the song Happy (for the song Eaten, $\bar{X} = 0.44$, s.e. $= 0.03$; for the song Happy, $\bar{X} = 0.37$, s.e. $= 0.02$; $F_{1,47} = 9.67$, $p = 0.005$, $\eta^2_G = 0.048$). By contrast, no such difference was observed for the fan group (for the song Eaten, $\bar{X} = 0.41$, s.e. $= 0.03$; for the song Happy, $\bar{X} = 0.42$, s.e. $= 0.03$; $p > 0.5$; figure 2a). Furthermore, as shown in figure 2b, the initial percept of neutral pictures was not influenced by the music types either for the non-fan group (for the song Eaten, $\bar{X} = 0.29$, s.e. $= 0.02$; for the song Happy, $\bar{X} = 0.32$, s.e. $= 0.02$; $p > 0.1$) or the fan group (for the song Eaten, $\bar{X} = 0.32$, s.e. $= 0.03$; for the song Happy, $\bar{X} = 0.30$, s.e. $= 0.03$; $p > 0.4$).

For the index of the mean percept duration, the repeated-measures ANOVA revealed a significant main effect of the Picture ($F_{1,78} = 19.12$, $p < 0.001$, $\eta^2_G = 0.054$). There were no other significant effects ($ps > 0.1$). This finding indicates that fans and non-fans of violent music perceived the violent pictures longer than the neutral pictures regardless of the music (fans: for violent pictures,

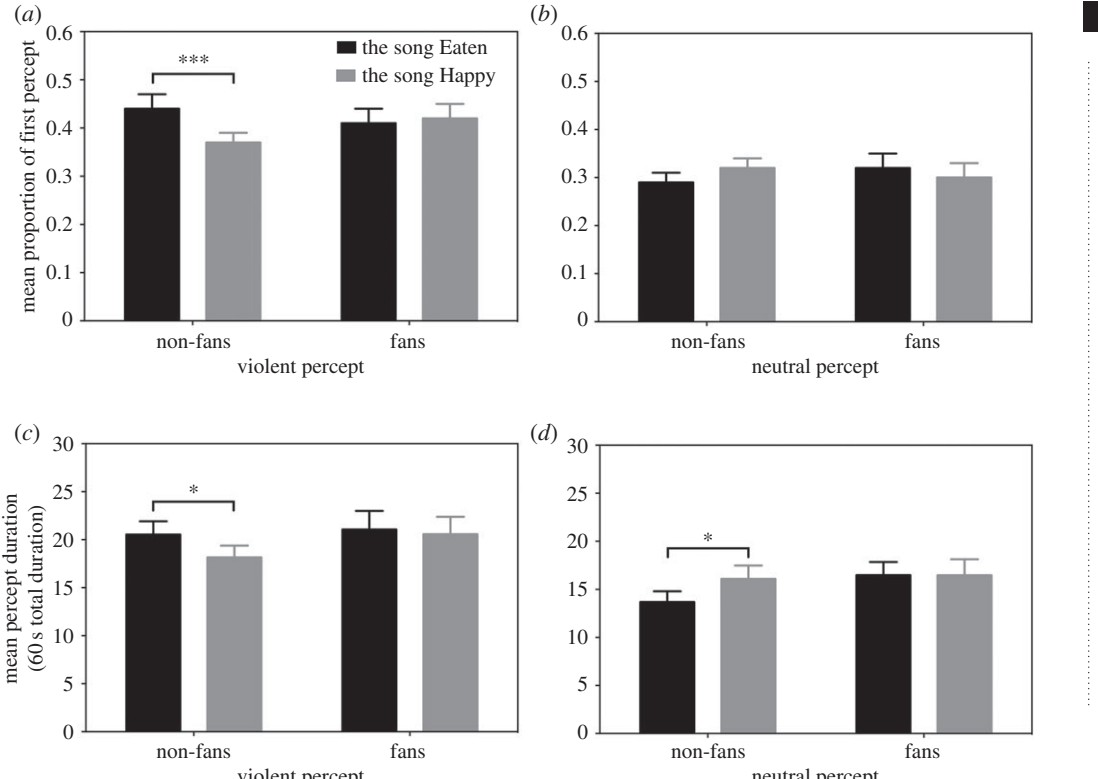

**Figure 2.** (*a*) Mean proportion of 16 trials for which participants' first percept was the violent picture. (*b*) Mean proportion of 16 trials for which participants' first percept was the neutral picture. (*c*) Mean cumulative time (in seconds) of perceiving violent pictures. (*d*) Mean cumulative time (in seconds) of perceiving neutral pictures. Error bar represents 1 standard error of the mean. *$p < 0.05$; ***$p < 0.005$.

$\bar{X} = 19.36$ s, s.e. $= 1.27$ s; for neutral pictures, $\bar{X} = 14.89$ s, s.e. $= 1.11$ s; non-fans: for violent pictures, $\bar{X} = 20.82$ s, s.e. $= 1.56$ s; for neutral pictures, $\bar{X} = 16.48$ s, s.e. $= 1.36$ s).

We next examined the mean durations of violent percept for the violent and happy music conditions in the non-fan and fan groups. As shown in figure 2*c*, non-fans perceived violent pictures significantly longer while listening to the song Eaten than while listening to the song Happy (for the song Eaten, $\bar{X} = 20.55$ s, s.e. $= 1.36$ s; for the song Happy, $\bar{X} = 18.16$ s, s.e. $= 1.23$ s; $F_{1,47} = 4.58$, $p = 0.038$, $\eta_G^2 = 0.018$). By contrast, for fans, the mean durations of the violent percept were similar in the two music conditions (for the song Eaten, $\bar{X} = 21.07$ s, s.e. $= 1.92$ s; for the song Happy, $\bar{X} = 20.57$ s, s.e. $= 1.82$ s; $p > 0.7$).

We next compared the mean durations of neutral percept for the violent and happy music conditions in each group. Non-fans perceived neutral pictures for longer while listening to the song Happy than while listening to the song Eaten (for the song Happy, $\bar{X} = 16.09$ s, s.e. $= 1.39$ s; for the song Eaten, $\bar{X} = 13.69$ s, s.e. $= 1.11$ s; $F_{1,47} = 4.56$, $p = 0.038$, $\eta_G^2 = 0.019$). By contrast, fans did not exhibit a significant difference in the duration of the neutral percept for the two music conditions (for the song Happy, $\bar{X} = 16.47$ s, s.e. $= 1.66$ s; for the song Eaten, $\bar{X} = 16.48$ s, s.e. $= 1.37$ s; $p > 0.9$; figure 2*d*).

An analysis of valence ratings for the songs Eaten and Happy revealed a significant effect of Music ($F_{1,78} = 35.76$, $p < 0.001$, $\eta_G^2 = 0.202$), a significant effect of Group ($F_{1,78} = 3.97$, $p = 0.050$, $\eta_G^2 = 0.022$) and a significant interaction between Music and Group ($F_{1,78} = 29.42$, $p < 0.001$, $\eta_G^2 = 0.172$). A follow-up simple effect test revealed that non-fans assigned significantly higher ratings of valence for the song Happy ($\bar{X} = 5.21$, s.e. $= 0.26$) than for the song Eaten ($\bar{X} = 2.00$, s.e. $= 0.22$; $F_{1,47} = 90.04$, $p < 0.001$, $\eta_G^2 = 0.499$). By contrast, fans assigned similar ratings of valence to the song Eaten ($\bar{X} = 4.19$, s.e. $= 0.33$) and the song Happy ($\bar{X} = 4.03$, s.e. $= 0.27$; $p > 0.7$; figure 3*a*).

The result of arousal ratings for the two songs revealed a significant interaction between Music and Group ($F_{1,78} = 8.04$, $p = 0.006$, $\eta_G^2 = 0.040$). The main effect of Group and the main effect of Music were not significant ($ps > 0.6$), suggesting that across participants, the two pieces of music were perceived to have similar levels of arousal. Further analysis on the interaction between Music and Group showed that non-fans assigned significantly higher ratings of arousal to the song Happy ($\bar{X} = 3.91$, s.e. $= 0.22$) than to the song Eaten ($\bar{X} = 3.12$, s.e. $= 0.27$; $F_{1,47} = 4.56$, $p = 0.038$, $\eta_G^2 = 0.041$). By contrast, fans assigned

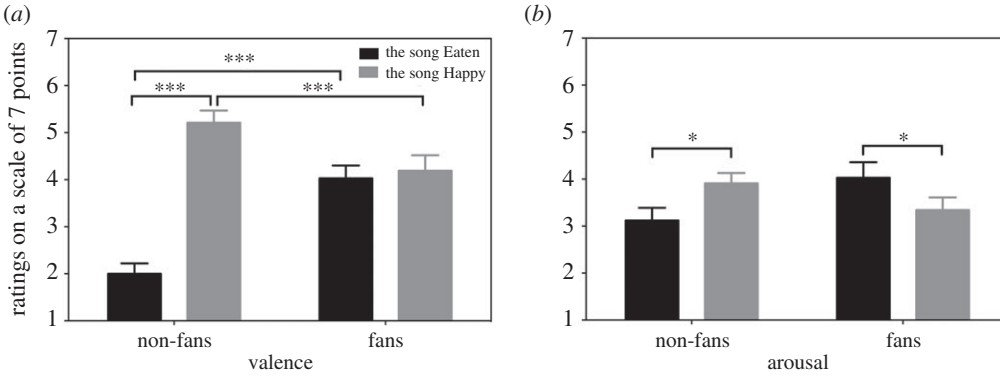

**Figure 3.** (a) Ratings on valence of the songs Eaten and Happy for each group. (b) Ratings on arousal of the songs Eaten and Happy for each group. Error bar represents 1 standard error of the mean. *$p < 0.05$; ***$p < 0.005$.

significantly higher ratings of arousal to the song Eaten ($\bar{X} = 4.03$, s.e. $= 0.33$) than to the song Happy ($\bar{X} = 3.34$, s.e. $= 0.27$; $F_{1,31} = 4.15$, $p = 0.050$, $\eta_G^2 = 0.046$; figure 3b).

## 4. Discussion

The results of this investigation confirm that both fans and non-fans of violent music exhibit a reliable bias for processing violent imagery over neutral imagery regardless of what genres of music they were listening to. Thus, we observed no evidence that fans of violent music are generally desensitized to violence, in that such desensitization should have been reflected in the absence of a bias for processing violent imagery for fans of violent music. Consistent with the negativity bias reported for the general population, fans of violent music exhibited a robust bias for processing violent imagery over non-violent imagery. This finding raises doubts about arguments that long-term exposure to violent media may desensitize consumers to violence [69–71]. Desensitization has been measured in different ways, but the binocular rivalry paradigm is uniquely powerful in that it provides a robust index of a subcortical processing bias for violent imagery and any reduction in such bias for fans of violent music. That fans did not exhibit reduced bias for processing violent images implies that the consequences of long-term exposure to violent music may be minimal or restricted to conscious levels of processing.

Indeed, recent research suggests that fans and non-fans of death metal music exhibit similar empathic capability [43], which raises doubts about the grave concerns that have been voiced about the dangers of exposure to violent music. Quite possibly, fans are able to frame their engagement with death metal music as an aesthetic experience that can be distinguished from real-life circumstances of violence [43]. Another possibility is that music with violent themes does not tend to express a clear or coherent narrative, making it unlikely that fans would contemplate how the violent themes expressed in the music might relate to real-life circumstances [5].

Our findings corroborate previous reports of a bias to perceive negative information over neutral or positive information in binocular rivalry paradigms [39,72]. Neuroscientific studies have also shown increased neural response to stimuli they deem negative [73,74]. Generally, humans are reflexively vigilant to negative information [75,76]. This bias has been interpreted as reflecting an adaptive instinct to prioritize negative information, given that organisms that are inclined to process negative conditions may have been more likely to survive in hostile environments [77].

Fans and non-fans of violent music did respond differently, however, to the presence of music. The implicit processing of violent and neutral imagery, as reflected in binocular rivalry, was significantly affected by the songs Eaten and Happy for non-fans of violent music, but no such effect was observed for fans of this music. For non-fans, a processing bias for violent imagery over neutral imagery was reinforced while listening to the song Eaten and attenuated while listening to the song Happy. This effect probably occurred because the two songs differed in their emotional effects on listeners. For non-fans of violent music, the song Happy primed them to perceive *neutral imagery* more frequently and longer, whereas the song Eaten primed them to perceive *violent imagery* more frequently and longer. By contrast, fans of violent music exhibited similar performance across these two music conditions. These findings may reflect the affective consequences of listening to the songs

Happy and Eaten. As the rating task revealed, compared to the song Happy, the song Eaten left non-fans feeling less pleasant (low ratings of valence) and less aroused (low ratings of arousal). By contrast, fans assigned similar ratings of valence to the two songs, and higher ratings of arousal to the song Eaten than to the song Happy.

Fans of violent music were unaffected by background music, as the bias for processing violent imagery was equivalent for the songs Happy and Eaten, even though the song Eaten has explicitly violent lyrics and is a prototypical example of music that the fans themselves interpret as expressing violent and aggressive themes. This outcome probably reflects the fact that fans of heavy and death metal did not perceive the song Eaten to differ from the song Happy in affective valence, with both songs assigned high mean ratings. Previous research has found that fans of violent 'death metal' music experience power, joy and peace while listening to this music [43]. Therefore, fans may have experienced similar affective states while listening to the songs Eaten and Happy, such that the two genres of music had a similar impact on binocular rivalry.

Thus, differences in the emotional experiences of fans and non-fans to the two songs may account for differences in the effect of music on binocular rivalry. However, it should be acknowledged that fans and non-fans differed in other ways. Fans of extreme metal music, by definition, enjoy this music and hence seek out opportunities to expose themselves to it. This high exposure, in turn, leads to increased familiarity that further contributes to enjoyment [78,79]. Thus, fans and non-fans differed from each other not only in their emotional responses to the song Eaten, but also in their level of enjoyment and exposure to this genre of music, and their familiarity with it. Given the continuous, long-term mutual reinforcement of enjoyment, exposure and familiarity, it is difficult to evaluate the independent importance of these variables.

In contrast with fans, exposure to music had a significant impact on the negativity bias for non-fans of violent music. For non-fans, a processing bias for violent imagery over neutral imagery was reinforced while listening to the song Eaten and attenuated while listening to the song Happy. This finding may reflect the contrasting affective states that were induced by the two songs. Based on the affective ratings, the song Eaten induced a negative affective state in non-fans, which may have led them to become more sensitive to violent imagery in accordance with the 'negative congruence effect' [39]. Conversely, a 'positive congruence effect' in non-fans was observed with the song Happy. Non-fans reported that they experienced pleasant feelings from the song Happy and, in turn, they exhibited greater subconscious processing of neutral imagery relative to violent imagery. Consistent with previous studies (e.g. [39,80,81]), this finding indicates that top-down processing can influence the contents of the subconscious processes that are associated with binocular rivalry. One implication of this finding is that short-term exposure to music that has positive connotations within a culture may distract them from the negative or violent information, and hence may be a useful therapeutic strategy for individuals who tend to dwell on violent imagery, such as sufferers of post-traumatic stress disorder.

Differences between fans and non-fans in the way that music influenced implicit processing of violent imagery illustrate the importance of individual differences in how people derived meaning and experience from music [82,83]. Fans and non-fans have contrasting affective experiences while listening to music, and such differences reflect individual circumstances, personality, and social and cultural influences. Although our investigation considered music in which violent lyrics are accompanied by aggressive musical sounds, it should be noted that lyrics and music need not always have congruent emotional connotations. For example, the lyrics of 'John Wayne Gacy Jr' by Sufjan Stevens depicts a serial killer, but the music is calm and relaxing. Similarly, 'Strange Fruit' by Billy Holiday describes the lynching of African Americans, but the music has ambiguous connotations that could even be construed as peaceful for some listeners. The relative importance of lyrics and sounds of music would be a valuable topic for future research.

## 5. Conclusion

The results of the binocular rivalry task suggest that long-term exposure to music with aggressive themes does not lead to a general desensitization to violence as depicted in images. Fans of violent music, just like non-fans, showed a robust bias to process violent imagery. However, results also suggest that fans of violent music do become desensitized to the aggressive and violent themes in their preferred music, in that they assigned high ratings of emotional valence relative to those of non-fans. High valence ratings suggest that fans derived positive emotional experiences from the music, which were probably reinforced by high levels of exposure and familiarity to the music. Desensitization to violence in music was also

illustrated in the differential effects of music on the bias for processing violent imagery, observed in the binocular rivalry task. For non-fans, the presence of music significantly influenced the bias for processing violent imagery, with aggressively themed music associated with increased bias, and positively themed music associated with decreased bias. However, no such effect of music was observed for fans of violent music, presumably because they experienced the two types of music as similarly positive. Taken together, these findings suggest that implicit processing of violent imagery is significantly influenced by the affective experience induced by music, but not by the perception of violent or aggressive themes in music. For listeners who extract a positive experience from violent or aggressively themed music—even when they recognize that the music expresses violence—music will not reinforce a processing bias for violent imagery any more than a positively themed song such as 'Happy'.

Ethics. All participants provided their written informed consent and received payments or course credit points for their time. This study was carried out in accordance with approved guidelines of the National Statement on Ethical Conduct in Human Research (2007). Ethical approval was obtained from the Ethics Committee of Macquarie University (reference no. 5201600580).

Data accessibility. The dataset supporting this article has been deposited at the Dryad Digital Repository: http://dx.doi.org/10.5061/dryad.kq487nj [84].

Authors' contributions. W.F.T., X.L. and M.W. conceived and designed the experiment. X.L. programmed the binocular rivalry task. Y.S. and X.L. collected the data. Y.S. analysed the data. Y.S. and X.L. wrote up the first draft of the manuscript. Y.S., W.F.T. and M.W. interpreted the data. Y.S. and X.L. contributed equally to the research.

Competing interests. We declare we have no competing interests.

Funding. This research was supported by an Australian Research Council Discovery Project grant no. (DP160101470) awarded to W.F.T.

Acknowledgements. We thank members of the Macquarie University Music, Sound and Performance Research Group for helpful comments on an earlier draft. We also thank Dr Fei Yi for her assistance in coding the experiment.

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
