## [Reviewer comments · Royal Society Open Science]

Review History

RSOS-181580.R0 (Original submission)

Review form: Reviewer 1 (Jonna Vuoskoski)

Is the manuscript scientifically sound in its present form?

Yes

Are the interpretations and conclusions justified by the results?

Yes

Is the language acceptable?

Yes

Is it clear how to access all supporting data?

Yes

Do you have any ethical concerns with this paper?

No

Have you any concerns about statistical analyses in this paper?

Yes

Recommendation?

Accept with minor revision (please list in comments)

Comments to the Author(s)

This manuscript reports an interesting study that investigated the effects of different kinds of background music (happy vs. violent) on a binocular rivalry task (with neutral vs. violent images), and how these effects might depend on the participants' music preferences/long-term music exposure. Overall, I find the study well designed and written, and my only real concern is related to familiarity and its potential role in some of the observed effects. With familiarity I mean both veridical familiarity with the specific violent and happy pieces used in the experiment, as well as stylistic familiarity with the respective musical genres. It is likely that the fans and non-fans of violent music differed in terms of both veridical and stylistic familiarity with the music pieces/genres used in the experiment. Was familiarity measured at all? Familiarity is known to increase liking for music (see e.g., North & Hargreaves, 1995; Peretz et al., 1998), and thus the differences in the valence ratings of the fans and non-fans of violent music might reflect differences in familiarity rather than differential emotional processing of the music. This possibility should be taken into consideration and discussed, and some of the conclusions (e.g., that "fans of violent music are desensitized to the aggressive and violent themes within this music", p. 10, rows 51-52) adjusted accordingly.

Some minor points:

- Page 5, description of procedure (rows 46-49): "each violent-neutral pair was presented twice, where the violent was present to the left eye in one trial and to the right eye in the repeated trial." Were these orders randomized or counterbalanced somehow? Or were all violent images always presented to the left eye first? Please clarify.
- Reporting effect sizes in the ANOVA analyses: I would advice you to report generalized eta squared (see Bakeman, 2005) instead of partial eta squared, since it provides comparability across between-subjects and within-subjects effects.
- Page 8, rows 27-19: The F-statistics and effect sizes are identical for the main effect of Group and for the interaction effect of Group & Music. Is this a typo?

References:

Bakeman, R. (2005). Recommended effect size statistics for repeated measures designs. *Behavior Research Methods*, 37(3), 379-384.

North, A. C., & Hargreaves, D. J. (1995). Subjective complexity, familiarity, and liking for popular music. *Psychomusicology: A Journal of Research in Music Cognition*, 14(1-2), 77.

Peretz, I., Gaudreau, D., & Bonnel, A. M. (1998). Exposure effects on music preference and recognition. *Memory & Cognition*, 26(5), 884-902.

Review form: Reviewer 2

Is the manuscript scientifically sound in its present form?

Yes

Are the interpretations and conclusions justified by the results?

No

Is the language acceptable?

Yes

Is it clear how to access all supporting data?

Yes

Do you have any ethical concerns with this paper?

No

Have you any concerns about statistical analyses in this paper?

I do not feel qualified to assess the statistics

Recommendation?

Major revision is needed (please make suggestions in comments)

Comments to the Author(s)

This paper is based on an interesting study but there are far too many assumptions and bias involved that invalidates the results. It is assumed that certain songs are inherently violent or happy and it is not discussed how these assumptions were arrived at. Social and cultural influences and personal experiences on how meaning is attached to musical phenomenon are ignored. Music has no inherent meaning and meaning of any particular recording often changes over time depending on memories, emotions and beliefs associated with the music, the context and other socio-cultural factors. In addition, it is not clear how violence and happiness was defined here: was it the music or the lyrics? Or both? Unclear.

More specifically:

P2 L21-2: The abstract mentions that it was suggested that violent media influences levels of violent behaviour, but there are no references to demonstrate this. The authors have not defined what they mean by happy and violent music nor how they came to these definitions. They have ignored their own cultural bias when determining this.

P2 L28-52: It is reductionist to claim that all death and heavy metal lyrics are violent. It is also reductionist to claim that heavy metal and death metal are the same in terms of sonic world and lyrical content. The correlational studies mentioned have not demonstrated any direct relationship between the music and the behaviour for the same reason that the cultural and social influences were not investigated.

P4 L35-40: The song choices are arbitrary and the definitions of the groups (fans and non-fans) poorly defined. Who has the power to make these decisions? How has researcher-bias influenced these decisions? What were the musical histories of the participants and how has that influenced their musical choices? Do fans of 'violent' music feel 'happy' when they listen to it? If so, then that is their happy music. Emotions are attached to music through experience, they are not inherent. Also, tools such as the MIRtoolbox do not enable any understanding of how meaning is attached to auditory experiences. In other words, breaking down the mechanics of a piece of music in this way ignores how social and cultural processes inform emotions, memories and beliefs about music and the results are unlikely to yield anything of use.

P11 L38-45: The authors imply that there is an object known as happy music that when applied to people who focus on violence, will distract them away from such violence. No musical experience

has a causal effect that can be measured and predicted in this way, since the meaning attached to the experience is largely formed through emotions, memories and beliefs about the musical phenomenon, the current situation and any situations associated with the music in the past. In general, it is very vague what exactly the researchers are discussing. They say they found similar physical elements of sound in the two song examples yet they discuss violent themes, which indicates that they are discussing lyrics, and nothing really to do with the sound at all. Without understanding what musical experiences, including and not-including music, means to groups and how those meanings were established, then the exposure of any particular music at a particular time in a particular context will not be able to determine why the results occurred. In other words, without a sociological grounding this form of research on musical impact is deeply flawed.

Some further general comments:

- There is a focus on death metal, yet the authors refer to violence in hip hop lyrics in their references, but this is not explored at all in the article.
- The heavy metal papers referenced are mostly from the 1990s and refer to music in the 1980s when the dominant narrative linked such music to deviant behaviour. These studies do not consider socio-cultural influences on behaviour and the resultant association to musical phenomenon either.
- This paper also does not consider examples where music might be experienced as happy but the lyrics are violent in nature, or vice versa. For example:

Examples:

“Ding dong the witch is dead” Wizard of Oz. Violent lyrics, jaunty music, happy imagery.
 Robin Thicke’s “Blurred Lines” and rape. Popular song in the style of Pharrel Williams.
 Sufjan Stevens “John Wayne Gacy Jr” about a serial killer, very calm and relaxed style of music.
 Plenty of metal songs with positive lyrics as well, even if they do tend to focus on the darker more melancholic side of human experience, but without that there would be no validity amongst the group membership who values the sharing of those experiences.

- Rainbow in the Dark – Dio

Or stories from literature

- Rime of the Ancient Mariner – Iron Maiden

Or science fiction

- Voivod – Lost Machine

Decision letter (RSOS-181580.R0)

17-Dec-2018

Dear Dr Sun,

The editors assigned to your paper ("Implicit Violent Imagery Processing among Fans and Non-fans of Violent Music") have now received comments from reviewers. We would like you to revise your paper in accordance with the referee and Associate Editor suggestions which can be found below (not including confidential reports to the Editor). Please note this decision does not guarantee eventual acceptance.

Please submit a copy of your revised paper before 09-Jan-2019. Please note that the revision deadline will expire at 00.00am on this date. If we do not hear from you within this time then it will be assumed that the paper has been withdrawn. In exceptional circumstances, extensions may be possible if agreed with the Editorial Office in advance. We do not allow multiple rounds of revision so we urge you to make every effort to fully address all of the comments at this stage.

If deemed necessary by the Editors, your manuscript will be sent back to one or more of the original reviewers for assessment. If the original reviewers are not available, we may invite new reviewers.

- Data accessibility

If you wish to submit your supporting data or code to Dryad (<http://datadryad.org/>), or modify your current submission to dryad, please use the following link:
<http://datadryad.org/submit?journalID=RSOS&manu=RSOS-181580>

- Competing interests

- Authors' contributions

- Acknowledgements

- Funding statement

Kind regards,

Andrew Dunn

on behalf of Dr Peter Keller (Associate Editor) and Antonia Hamilton (Subject Editor)

Associate Editor's comments (Dr Peter Keller):

Two experts have reviewed your manuscript and both found the study to be interesting. However, Reviewer 1 raised concerns about the issue of familiarity with the musical materials and recommended minor revisions to address this issue. Reviewer 2 has serious reservations about the validity of the research with respect to issues related to musical meaning and the definition of key constructs such as violence and happiness. Major revision aimed at clarifying relevant assumptions would be needed to address this reviewer's concerns.

Comments to Author:

Reviewers' Comments to Author:

Reviewer: 1

Comments to the Author(s)

This manuscript reports an interesting study that investigated the effects of different kinds of background music (happy vs. violent) on a binocular rivalry task (with neutral vs. violent images), and how these effects might depend on the participants' music preferences/long-term music exposure. Overall, I find the study well designed and written, and my only real concern is related to familiarity and its potential role in some of the observed effects. With familiarity I mean both veridical familiarity with the specific violent and happy pieces used in the experiment, as well as stylistic familiarity with the respective musical genres. It is likely that the fans and non-fans of violent music differed in terms of both veridical and stylistic familiarity with the music pieces/genres used in the experiment. Was familiarity measured at all? Familiarity is known to

increase liking for music (see e.g., North & Hargreaves, 1995; Peretz et al., 1998), and thus the differences in the valence ratings of the fans and non-fans of violent music might reflect differences in familiarity rather than differential emotional processing of the music. This possibility should be taken into consideration and discussed, and some of the conclusions (e.g., that “fans of violent music are desensitized to the aggressive and violent themes within this music”, p. 10, rows 51-52) adjusted accordingly.

Some minor points:

- Page 5, description of procedure (rows 46-49): “each violent-neutral pair was presented twice, where the violent was present to the left eye in one trial and to the right eye in the repeated trial.” Were these orders randomized or counterbalanced somehow? Or were all violent images always presented to the left eye first? Please clarify.

- Reporting effect sizes in the ANOVA analyses: I would advice you to report generalized eta squared (see Bakeman, 2005) instead of partial eta squared, since it provides comparability across between-subjects and within-subjects effects.

- Page 8, rows 27-19: The F-statistics and effect sizes are identical for the main effect of Group and for the interaction effect of Group & Music. Is this a typo?

References:

Bakeman, R. (2005). Recommended effect size statistics for repeated measures designs. *Behavior Research Methods*, 37(3), 379-384.

North, A. C., & Hargreaves, D. J. (1995). Subjective complexity, familiarity, and liking for popular music. *Psychomusicology: A Journal of Research in Music Cognition*, 14(1-2), 77.

Peretz, I., Gaudreau, D., & Bonnel, A. M. (1998). Exposure effects on music preference and recognition. *Memory & Cognition*, 26(5), 884-902.

Reviewer: 2

Comments to the Author(s)

This paper is based on an interesting study but there are far too many assumptions and bias involved that invalidates the results. It is assumed that certain songs are inherently violent or happy and it is not discussed how these assumptions were arrived at. Social and cultural influences and personal experiences on how meaning is attached to musical phenomenon are ignored. Music has no inherent meaning and meaning of any particular recording often changes over time depending on memories, emotions and beliefs associated with the music, the context and other socio-cultural factors. In addition, it is not clear how violence and happiness was defined here: was it the music or the lyrics? Or both? Unclear.

More specifically:

P2 L21-2: The abstract mentions that it was suggested that violent media influences levels of violent behaviour, but there are no references to demonstrate this. The authors have not defined what they mean by happy and violent music nor how they came to these definitions. They have ignored their own cultural bias when determining this.

P2 L28-52: It is reductionist to claim that all death and heavy metal lyrics are violent. It is also reductionist to claim that heavy metal and death metal are the same in terms of sonic world and

lyrical content. The correlational studies mentioned have not demonstrated any direct relationship between the music and the behaviour for the same reason that the cultural and social influences were not investigated.

P4 L35-40: The song choices are arbitrary and the definitions of the groups (fans and non-fans) poorly defined. Who has the power to make these decisions? How has researcher-bias influenced these decisions? What were the musical histories of the participants and how has that influenced their musical choices? Do fans of 'violent' music feel 'happy' when they listen to it? If so, then that is their happy music. Emotions are attached to music through experience, they are not inherent. Also, tools such as the MIRtoolbox do not enable any understanding of how meaning is attached to auditory experiences. In other words, breaking down the mechanics of a piece of music in this way ignores how social and cultural processes inform emotions, memories and beliefs about music and the results are unlikely to yield anything of use.

P11 L38-45: The authors imply that there is an object known as happy music that when applied to people who focus on violence, will distract them away from such violence. No musical experience has a causal effect that can be measured and predicted in this way, since the meaning attached to the experience is largely formed through emotions, memories and beliefs about the musical phenomenon, the current situation and any situations associated with the music in the past. In general, it is very vague what exactly the researchers are discussing. They say they found similar physical elements of sound in the two song examples yet they discuss violent themes, which indicates that they are discussing lyrics, and nothing really to do with the sound at all. Without understanding what musical experiences, including and not-including music, means to groups and how those meanings were established, then the exposure of any particular music at a particular time in a particular context will not be able to determine why the results occurred. In other words, without a sociological grounding this form of research on musical impact is deeply flawed.

Some further general comments:

- There is a focus on death metal, yet the authors refer to violence in hip hop lyrics in their references, but this is not explored at all in the article.
- The heavy metal papers referenced are mostly from the 1990s and refer to music in the 1980s when the dominant narrative linked such music to deviant behaviour. These studies do not consider socio-cultural influences on behaviour and the resultant association to musical phenomenon either.
- This paper also does not consider examples where music might be experienced as happy but the lyrics are violent in nature, or vice versa. For example:

Examples:

“Ding dong the witch is dead” Wizard of Oz. Violent lyrics, jaunty music, happy imagery.

Robin Thicke’s “Blurred Lines” and rape. Popular song in the style of Pharrel Williams.

Sufjan Stevens “John Wayne Gacy Jr” about a serial killer, very calm and relaxed style of music.

Plenty of metal songs with positive lyrics as well, even if they do tend to focus on the darker more melancholic side of human experience, but without that there would be no validity amongst the group membership who values the sharing of those experiences.

- Rainbow in the Dark – Dio

Or stories from literature

- Rime of the Ancient Mariner – Iron Maiden

Or science fiction

- Voivod – Lost Machine

Author's Response to Decision Letter for (RSOS-181580.R0)

See Appendix A.

RSOS-181580.R1 (Revision)

Review form: Reviewer 2

Is the manuscript scientifically sound in its present form?

Yes

Are the interpretations and conclusions justified by the results?

Yes

Is the language acceptable?

Yes

Is it clear how to access all supporting data?

Not Applicable

Do you have any ethical concerns with this paper?

No

Have you any concerns about statistical analyses in this paper?

I do not feel qualified to assess the statistics

Recommendation?

Accept as is

Comments to the Author(s)

You have addressed my concerns, more or less, so thanks for your diligence. I remain sceptical about the usefulness of this approach to understanding the impact of music on behaviour, but within the field that it is situated, it is certainly publishable. Perhaps it will start a further debate!

Decision letter (RSOS-181580.R1)

12-Feb-2019

Dear Dr Sun,

I am pleased to inform you that your manuscript entitled "Implicit Violent Imagery Processing among Fans and Non-fans of Violent Music" is now accepted for publication in Royal Society Open Science.

Royal Society Open Science operates under a continuous publication model (<http://bit.ly/cpFAQ>). Your article will be published straight into the next open issue and this

will be the final version of the paper. As such, it can be cited immediately by other researchers. As the issue version of your paper will be the only version to be published I would advise you to check your proofs thoroughly as changes cannot be made once the paper is published.

on behalf of Dr Peter Keller (Associate Editor) and Antonia Hamilton (Subject Editor)
openscience@royalsociety.org

Associate Editor Comments to Author (Dr Peter Keller):

Associate Editor: 1

Comments to the Author:

One of the original reviewers evaluated your revised manuscript. Although they question the general approach, they recognise the value of adopting different perspectives to stimulate further debate, and recommend publication. Given the clear and compelling research question, and technical soundness of the research, I agree with the evaluation and see the potential for cross-discipline exchange as highly fruitful.

Reviewer comments to Author:

Reviewer: 2

Comments to the Author(s)

You have addressed my concerns, more or less, so thanks for your diligence. I remain sceptical about the usefulness of this approach to understanding the impact of music on behaviour, but within the field that it is situated, it is certainly publishable. Perhaps it will start a further debate!

Appendix A

Reviewer: 1

Comments to the Author(s)

This manuscript reports an interesting study that investigated the effects of different kinds of background music (happy vs. violent) on a binocular rivalry task (with neutral vs. violent images), and how these effects might depend on the participants' music preferences/long-term music exposure. Overall, I find the study well designed and written, and my only real concern is related to familiarity and its potential role in some of the observed effects. With familiarity I mean both veridical familiarity with the specific violent and happy pieces used in the experiment, as well as stylistic familiarity with the respective musical genres. It is likely that the fans and non-fans of violent music differed in terms of both veridical and stylistic familiarity with the music pieces/genres used in the experiment. Was familiarity measured at all? Familiarity is known to increase liking for music (see e.g., North & Hargreaves, 1995; Peretz et al., 1998), and thus the differences in the valence ratings of the fans and non-fans of violent music might reflect differences in familiarity rather than differential emotional processing of the music. This possibility should be taken into consideration and discussed, and some of the conclusions (e.g., that "fans of violent music are desensitized to the aggressive and violent themes within this music", p. 10, rows 51-52) adjusted accordingly.

We thank the reviewer for the comment. We have acknowledged this possibility in the discussion section as "This high exposure, in turn, leads to increased familiarity that further contributes to enjoyment [78, 79]. Thus, fans and non-fans differed from each other not only in their emotional responses to the song Eaten, but also in their level of enjoyment and exposure to this genre of music, and their familiarity with it." (See Page 5 Line 60 and Page 6 Line 1-3.)

We've also modified the conclusion accordingly as "High valence ratings suggest that fans derived positive emotional experiences from the music, which were likely reinforced by high levels of exposure and familiarity to the music." (See Page 6 Line 32-33.)

Some minor points:

- Page 5, description of procedure (rows 46-49): "each violent-neutral pair

was presented twice, where the violent was present to the left eye in one trial and to the right eye in the repeated trial." Were these orders randomized or counterbalanced somehow? Or were all violent images always presented to the left eye first? Please clarify.

These orders were randomized. This has been clarified in Line 53-54 on Page 3 as "each violent-neutral pair was presented twice at random positions in the sequence of trials, where the violent was present to the left (right) eye in one trial and to the right (left) eye in the repeated trial."

- Reporting effect sizes in the ANOVA analyses: I would advise you to report generalized eta squared (see Bakeman, 2005) instead of partial eta squared, since it provides comparability across between-subjects and within-subjects effects.

Thanks for the suggestion. Generalized eta squared has been reported instead of partial eta squared across the whole article.

- Page 8, rows 27-19: The F-statistics and effect sizes are identical for the main effect of Group and for the interaction effect of Group & Music. Is this a typo?

Thanks for pointing out this. It is a typo. This has been corrected in the text. Please see Page 5 Line 9-10.

References:

Bakeman, R. (2005). Recommended effect size statistics for repeated measures designs. *Behavior Research Methods*, 37(3), 379-384.

North, A. C., & Hargreaves, D. J. (1995). Subjective complexity, familiarity, and liking for popular music. *Psychomusicology: A Journal of Research in Music Cognition*, 14(1-2), 77.

Peretz, I., Gaudreau, D., & Bonnel, A. M. (1998). Exposure effects on music preference and recognition. *Memory & Cognition*, 26(5), 884-902.

Thanks for the references. We have cited all of them in the article.

Reviewer: 2

Comments to the Author(s)

This paper is based on an interesting study but there are far too many assumptions and bias involved that invalidates the results. It is assumed that certain songs are inherently violent or happy and it is not discussed how these assumptions were arrived at. Social and cultural influences and personal experiences on how meaning is attached to musical phenomenon are ignored. Music has no inherent meaning and meaning of any particular recording often changes over time depending on memories, emotions and beliefs associated with the music, the context and other socio-cultural factors. In addition, it is not clear how violence and happiness was defined here: was it the music or the lyrics? Or both? Unclear.

We thank the reviewer for these comments. We are sympathetic to the points raised and have not made these assumptions, so infer that the manuscript needs clarification. We've addressed these points in three ways. First, we clarify the aim of our study, which was to examine implicit processing of violent imagery among self-declared fans and non-fans of music with violent themes. Our findings confirm what the reviewer asserts: that fans and non-fans derive different experiences from music, leading to differential effects of music on how violent images are processed. Second, we label songs by their titles, rather than as "violent music" or "happy music". Third, we clarify our use of the term violence, and adopt widely accepted criteria in the aggression and violence literature, and apply the terms to both the lyrics and music. For lyrics, depictions of violence in our song "Eaten" are unambiguous according to the World Health Organization definition. For non-lyrical qualities of music, we draw from two sources of evidence: 1) that emotion is expressed in music using the same "code" as that used in the human voice, which expresses emotionality effectively and unambiguously using prosodic signals that are readily decoded by listeners; 2) we reference the ethology literature on biological signals of aggression (e.g., growls, screams) and cite evidence that the acoustic qualities of "Eaten" (e.g., non-linearities, rapid pace) are congruent with aggressive sounds in nature. Please refer to Page 2 Line 28-34.

We take on board the reviewer's point that the emotional connotation of musical sounds is not rigid, and we are careful not to imply otherwise. To ensure no reader has the same concerns, we inserted a paragraph in the introduction that distinguishes between the complex way individuals derive meaning from a particular activity, and violence as a societal and legal

concept (refer to Page 2 Line 20-27). For example, fans of the game “Grand Theft Auto” typically experience joy and empowerment from the game; but society still considers murder – depicted repeatedly in the game – as an act of violence.

The current investigation addressed a very simple and fundamental question: Do fans of music with violent themes become desensitized to violence depicted visually? They do not. This finding will be surprising to many media-violence researchers, parent groups, and government policy makers, and yet it underscores exactly what the reviewer highlighted about the complex nature of musical meaning and experience.

More specifically:

P2 L21-2: The abstract mentions that it was suggested that violent media influences levels of violent behaviour, but there are no references to demonstrate this. The authors have not defined what they mean by happy and violent music nor how they came to these definitions. They have ignored their own cultural bias when determining this.

Response: The journal does not permit the use of references in the abstract. References demonstrating this point were amply cited in the introduction – see the first paragraph of the introduction.

The music used in the present experiment have been more clearly defined in the abstract. See Page 1 Line 33-34 as “..... while they heard Western popular music with lyrics that expressed happiness or Western extreme-metal music with lyrics that expressed violence.”

P2 L28-52: It is reductionist to claim that all death and heavy metal lyrics are violent. It is also reductionist to claim that heavy metal and death metal are the same in terms of sonic world and lyrical content. The correlational studies mentioned have not demonstrated any direct relationship between the music and the behaviour for the same reason that the cultural and social influences were not investigated.

Thanks for this comment. We agree not all metal lyrics are violent. We never made this claim and nor would we. Our research concerns people who are self-reported fans and frequent listeners of “violent music” which we defined clearly to participants. Therefore, it isn’t relevant that not all Death Metal music is violent, but we are now careful to make this point clear. We don’t want to overstate this point, because our research has determined that a

high proportion of death metal music does contain violent lyrics (as defined by the World Health Organization), but we now state explicitly in the text that not all death and heavy metal music contains violent lyrics as “Some genres of music, including but not limited to extreme metal, rap, and hip hop, are often, though certainly not always, characterised by aggressive sounds and violent lyrics.” (see Page 1 Line 51-52)

The reviewer makes a good point that our use of a forward slash between “heavy” and “death” could imply that these two categories are the same in terms of music and lyrical content. Thus, all instances of “death/heavy metal” have been changed to “death metal and heavy metal” across the article.

P4 L35-40: The song choices are arbitrary and the definitions of the groups (fans and non-fans) poorly defined. Who has the power to make these decisions? How has researcher-bias influenced these decisions? What were the musical histories of the participants and how has that influenced their musical choices? Do fans of 'violent' music feel 'happy' when they listen to it? If so, then that is their happy music. Emotions are attached to music through experience, they are not inherent.

We’ve provided a clearer justification for our song choices and additional details about participants in the two groups. We also make a clearer distinction between expressed and felt emotion. Please refer to Page 2 Line 20-50.

On the point that the songs are “arbitrary”, we disagree. We adopted predetermined criteria for selecting songs that were contrasting in their connotations. The song “Eaten” unambiguously depicts violence in the lyrics, and its sonic features express aggression according to ethological principles of biological signalling. In contrast, the song “Happy” depicts prosocial lyrics and has a conventional song structure written in a major key. It is true that other songs might have fulfilled these criteria, but methodological considerations limited the number of songs we could examine.

With respect to the reviewer’s question “Do fans of violent music feel happy when they listen to it?” – we are now clearer that our criteria for song choice did not include the emotional experience of participants while listening to the music (felt emotion), but was determined by the expression of violence and aggression in music (expressed emotion). Expression of violence and aggression was determined in two ways. First, violence was explicitly expressed in lyrics while the music contained features associated with

aggressive biological signals. Second, metal fans in our study explicitly indicated that they frequently listened to and enjoyed metal music “with violent themes”. Thus, we described the song “Eaten” as an instance of violent music based on its literal content, past empirical findings, ethological principles of biological signals, and how fans themselves construed this music.

The reviewer correctly notes that fans have positive “felt” experiences to this music. We agree: our valence ratings confirm this insight, as does our previous research on Death Metal music, which we cite in the manuscript.

Also, tools such as the MIRtoolbox do not enable any understanding of how meaning is attached to auditory experiences. In other words, breaking down the mechanics of a piece of music in this way ignores how social and cultural processes inform emotions, memories and beliefs about music and the results are unlikely to yield anything of use.

We address this comment in three ways. First, we clarify the aim of our research. Our aim was simple, and relevant to widespread concerns about the long-term effects of listening to music with violent themes: To determine how violent imagery is implicitly processed by individuals who frequently listen to and enjoy music with violent themes. Second, we now acknowledge in our discussion section that social and cultural processes “inform emotions, memories and beliefs about music” (see Page 6 Line 15-16). We never disagreed with this point, and indeed it helps to explain the differences that we observed in how music impacts upon implicit processing of violent imagery by fans and non-fans. Third, we clarify our reason for using the MIRtoolbox as to control the arousal of the songs arising from loudness and tempo, in order to reduce the potential impact of physiological arousal on binocular rivalry processing. It wasn’t used to investigate social and cultural processes – it’s simply a convenient and powerful way to measure and modify some acoustic features of music stimuli.

P11 L38-45: The authors imply that there is an object known as happy music that when applied to people who focus on violence, will distract them away from such violence. No musical experience has a causal effect that can be measured and predicted in this way, since the meaning attached to the experience is largely formed through emotions, memories and beliefs about the musical phenomenon, the current situation and any situations associated with the music in the past.

As we compared implicit violent imagery processing under experimental and comparison conditions, we can indeed infer that exposure to different types of music “caused” the pattern of effects observed for the non-fan group relative to the fan group. However, we acknowledge there are different interpretations of the effect. Please refer to Page 6 Line 4-14.

In general, it is very vague what exactly the researchers are discussing. They say they found similar physical elements of sound in the two song examples yet they discuss violent themes, which indicates that they are discussing lyrics, and nothing really to do with the sound at all. Without understanding what musical experiences, including and not-including music, means to groups and how those meanings were established, then the exposure of any particular music at a particular time in a particular context will not be able to determine why the results occurred. In other words, without a sociological grounding this form of research on musical impact is deeply flawed.

We thank the reviewer for picking up on this potential source of confusion, and we have reworded our descriptions accordingly. Briefly, we equated the two songs on overall loudness (intensity) and tempo, but not on other attributes. The songs are clearly very different in virtually all other respects, and we have described these differences in detail. Please see Page 3 Line 15-22.

We cannot determine the impact of musical sound versus the lyrics as we define the expressed emotion of music based on both musical sound and lyrics. Please refer to Page 2 Line 20-34, which states the definition of violent and happy music used in this study.

We have also labelled the two songs used in the experiment according to their names – “Eaten” and “Happy” rather than “violent music” and “happy music” throughout the article.

Again, our aim was to determine how violent images are implicitly processed by individuals who explicitly claim to enjoy music with violent themes. It was not to determine the impact of individual experience and socio-cultural factors or how the musical meanings were established.

Some further general comments:

- There is a focus on death metal, yet the authors refer to violence in hip hop lyrics in their references, but this is not explored at all in the article.

Although we tested fans of death metal or heavy metal music and the stimuli used in the experiment is also a death metal song, our focus is the violence in this kind of music rather the music genres per se. Therefore, we also included references to research on violence in non-metal music (including hip hop).

- The heavy metal papers referenced are mostly from the 1990s and refer to music in the 1980s when the dominant narrative linked such music to deviant behaviour. These studies do not consider socio-cultural influences on behaviour and the resultant association to musical phenomenon either.

We have 18 references to research that explicitly considers heavy or death metal, including four new references that we added in response to this comment. Most are recent and only 6 were published prior to 2001.

Brummert Lennings, H. I., Warburton, W. A. 2011 The effect of auditory versus visual violent media exposure on aggressive behaviour: The role of song lyrics, video clips and musical tone. J. Exp. Soc. Psychol. 47, 794-799. (doi:10.1016/j.jesp.2011.02.006)

Selfhout, M. H. W., Delsing, M. J. M. H., ter Bogt, T. F. M., Meeus, W. H. J. 2008 Heavy metal and hip-hop style preferences and externalizing problem behavior: A two-wave longitudinal study. Youth Soc. 39, 435-452. (doi:10.1177/0044118X07308069)

Shafron, G. R., Karno, M. P. 2013 Heavy metal music and emotional dysphoria among listeners. Psychol. Pop. Media Cult. 2, 74-85. (doi:10.1037/a0031722)

Sharman, L., Dingle, G. A. 2015 Extreme metal music and anger processing. Front. Hum. Neurosci. 9, 272-272. (doi:10.3389/fnhum.2015.00272)

- This paper also does not consider examples where music might be experienced as happy but the lyrics are violent in nature, or vice versa. For example:

Examples:

“Ding dong the witch is dead” Wizard of Oz. Violent lyrics, jaunty music, happy imagery.

Robin Thicke’s “Blurred Lines” and rape. Popular song in the style of Pharrel Williams.

Sufjan Stevens “John Wayne Gacy Jr” about a serial killer, very calm and relaxed style of music.

Plenty of metal songs with positive lyrics as well, even if they do tend to focus on the darker more melancholic side of human experience, but without that there would be no validity amongst the group membership who values the sharing of those experiences.

- Rainbow in the Dark – Dio

Or stories from literature

- Rime of the Ancient Mariner – Iron Maiden

Or science fiction

- Voivod – Lost Machine

We now note that future research may examine music with contrasting emotional connotations in the lyrics and music. Please refer to Page 6 Line 15-23.

We are well aware of the literature on the interaction between lyrics and song, and past research from our lab has made considerable progress in understanding how the many dimensions of music combine and interact in their psychological and cognitive-motor effects. Addressing such interactions was not our aim. Instead, our goal was to address fundamental questions that concern parents, the music industry, politicians, and government censorship policies: whether fans of metal music with explicitly violent lyrics are desensitised to violence, and whether listening to aggressive music influences implicit processing of violent imagery. We believe our results will have a significant impact on theory and research.

We acknowledge there are issues not addressed in our study, but no study in the psychology of music can be expected to consider every possible issue.